# The Power of Massage in Children with Cancer—How Can We Do Effective Research?

**DOI:** 10.3390/children6010013

**Published:** 2019-01-18

**Authors:** Shana Jacobs, Catriona Mowbray

**Affiliations:** Division of Oncology, Children’s National Medical Center, 111 Michigan Ave NW, Washington, DC 20010, USA; cmowbray@cnmc.org

**Keywords:** massage, integrative therapies, pediatric cancer

## Abstract

Children with cancer experience multiple troubling symptoms. Massage offers a safe, non-pharmacological approach to address these symptoms. Numerous studies of massage in children and adults with cancer have been performed, yet most are unable to demonstrate significant benefit. This review aims to summarize what we know about the role of massage and sets goals and challenges for future massage research. This paper descriptively reviews the existing literature available in PubMed (both prior reviews and select papers) and the holes in prior research studies. Prior research on massage has been limited by small sample size/insufficient power, inappropriate outcome measures or timing, heterogeneous patient populations, inconsistent intervention techniques, and other design flaws. Based on the findings and limitations of previous work, numerous suggestions are made for future studies to increase the potency of findings, including thoughts about appropriate dosing, control groups, type of intervention, outcome measures, patient selection, feasibility, and statistics.

## 1. Introduction

Children and adolescents with cancer experience multiple troubling and often inter-related symptoms, including disturbed sleep, fatigue, anxiety, and pain [1,2,3]. This report will review the evidence for using massage therapy in this population and offer recommendations for future constructive research in this promising field.

Decreased sleep efficiency and sleep disturbances are prevalent in children with cancer [4,5,6], and are especially troublesome in hospitalized patients [7,8], who may have up to 40 sleep interruptions per night [9] and poor sleep efficiency [10]. Disturbed sleep has negative health-related and psychological consequences [11,12].

Cancer-related fatigue is a well described though poorly understood multifactorial symptom described as an inability to function due to exhaustion that may be linked with poor sleep [13]. Fatigue is a frequent complaint for pediatric cancer patients compared to healthy peers [4,14]. Both poor sleep and fatigue are linked to multiple other distressing physical and psychological symptoms and poorer health-related quality of life (QOL) [9,14,15]. Fatigue is also more prevalent in pediatric patients with cancer who are hospitalized [4].

Pain and anxiety are also frequent complaints for pediatric cancer patients [2]. Pain can result from the cancer itself as well as the side effects of the treatment, including mucositis, constipation, and invasive procedures. As a result, patients routinely require high levels of opioids for pain relief with various degrees of success [16]. Similarly, anxiety is common in children and adolescents undergoing these treatments [17].

The treatments for fatigue are often ineffective or undesirable [18]. Treatments for pain and anxiety such as opioids and benzodiazepines are generally sedating and cause numerous side effects. The treatment for sleep disturbances and the psychological symptoms associated with poor sleep and fatigue often includes the prescription of psychotropic drugs, despite relatively little knowledge about using these medications in this population [19,20], and multiple drug interactions. Patients with cancer are prescribed multiple medications, which can lead to undesirable drug interactions, making non-pharmacological methods to address these troubling symptoms, such as massage, very appealing.

Use of integrative therapies, particularly among people with cancer and other serious illnesses, is quite common. Among children with cancer, many studies have reported high prevalence rates. Massage is a commonly employed therapy, and anecdotal reports consistently report positive experiences. In other words, most people feel better after massage. However, research studies have found it difficult to consistently document the positive effects of massage. In addition to summarizing what research has shown us about massage, this manuscript will also outline some of the challenges to researching massage therapy, as well as guide next steps for future research.

This paper does not aim to be a comprehensive systematic review. Rather, our goal was to descriptively analyze the existing literature and examine what the research has shown and where holes remain. To do this, we began by searching PubMed (database of US National Library of Medicine, National Institutes of Health) for all articles related to massage and cancer from 2000 to present. We reviewed all articles, and then described in the paper all relevant review articles and all articles that described well-done studies that were not included in prior reviews. We only included articles in English. Since our focus is on pediatric cancer, all articles that focused on massage in children with cancer were included and described in detail.

## 2. Massage in Adults with Cancer

Massage is a commonly employed supportive therapy in adults with cancer, and many small studies have been done evaluating its effects. Unfortunately, many of the studies have been underpowered, or have had design flaws that make them difficult to interpret. Several review articles have summarized the research to date.

Jane et al. [21] published a 2008 systematic review of 15 studies in English published from 1986–2006 related to massage interventions for adults with cancer. Most of the studies were conducted in the United States and were in a hospital setting. Most studies focused on patients on active therapy. Many studies had design flaws: 12 of 15 had small sample sizes, and only two had statistical power of at least 80%. Most had either no randomization or no control group or had only pre-/post test designs. In those with control groups, most did not consider covariates such as type of cancer or baseline pain, and most did not blind data collectors to the study assignment. There was considerable variation between intervention techniques (type of massage, body area included, dose and timing of massage), and only one described a standardized massage protocol. The most common outcomes included were: pain, as well as quality of life, fatigue, and psychological measures (anxiety, mood). Most used self-report measures such as visual analogue scales, while six also used physiologic parameters like heart rate. Most used only one dose and looked at pre–/post outcomes, and measured only short-term effects (up to 5 min). The one study that measured effects lasting up to 2 hours did not find sustained effects. Three studies also had qualitative measures, which were all positive. The authors concluded that despite large variability and design flaws, massage treatments appeared to have more positive effects than the control group in terms of decreasing pain intensity, nausea, fatigue, distressing symptoms, anxiety, and improving self-report of relaxation and physiological arousal (blood pressure) immediately or 5 or 10–20 min after massage, though not beyond 20 min after termination of massage. Other outcomes, such as sleep and quality of life, were more inconsistent. The authors recommended:Studies should be careful about inclusion criteria and confounders.Studies should include control groups and document homogeneity between groups.While a full double-blind randomization is not possible, those collecting data, such as those soliciting patient reported outcomes, should be blinded to treatment group.Research must identify the ideal type of massage, ideal time of day, and ideal dose.Researchers should select measurement instruments and timing of instruments carefully to be sure that the outcome measures are capable of detecting the desired outcome and that they are delivered at the proper time to adequately measure effects.Studies need sufficient numbers for improved statistical power.

Myers et al. [22] also published a review in 2008 for all articles up to 2007 that included patient-reported outcome measures (PROs) for massage therapy. Twenty-two studies in English were included: Swedish massage was the most frequent (11 studies), with five on aromatherapy, five on reflexology, and two on acupressure. Symptoms assessed included: anxiety (16), pain (15), nausea (9), depression (9), and fatigue (4). Measurement tools varied, with visual analogue scales being the most frequent. As with Jane [21], the authors documented tremendous variation in type of massage, massage provider experience, location of massage, types of patients, duration and frequency of intervention, and number of patients included. The authors concluded that there appears to be good evidence on the role of massage for reducing anxiety. They recommended:Future studies should include adequate control groups and between-group results, rather than just pre–/post results.Studies need sufficient numbers for improved statistical power.Studies require thoughtful decision related to dosing of massage and thoughtful consideration of timing of outcome measures; for example, perhaps anxiety is better measured immediately whereas sleep is a longer-term effect.

In 2015, Lee [23] published a meta-analysis of massage therapy on cancer pain. After reviewing all literature from 1990 to 2013, 12 studies were included. Studies evaluated body massage (7), foot reflexology (4), and aroma massage (2), and all used standard of care control. Most studies used multiple doses of massage over a period of weeks. The effect of massage therapy for patients with cancer pain compared to standard of care was significant based on a random-effects model meta-analysis of data from all 12 studies (SMD −1.25 (95% CI −1.63 to −0.87); *p* < 0.00001). The effects were significant regardless of reason for pain, type of cancer, and method and time point for assessing pain, although few studies looked at longer-term effects. The effects were also significant for all types of massage and for studies with high quality and lower quality methodology. The authors recommended:Measurement instruments should be standardized to allow for further meta-analyses.Standard of care control should be used as an appropriate comparator in massage studies, although sham Reiki techniques may also be used.

Shin et al. performed a Cochrane review in 2016 [24] of massage with or without aromatherapy for symptom relief in people with cancer. Nineteen studies were included; 14 were evaluated qualitatively and five were included in a meta-analysis. Overall, the analysis found all the studies to be of low quality evidence and found no statistically and clinically significant differences between massage versus no massage (with or without aromatherapy) for pain, anxiety, mood, fatigue, nausea, distress, or quality of life. The authors concluded: Intervention protocols should be standardized in terms of the number and duration of massage treatments (dose).The optimal massage techniques, the body parts to be massaged, and which essential oils should be blended into the carrier oil should be standardized.Research needs to identify reliable, validated tools to use for outcome measures.Large, well-designed studies are required to give some definitive answers to the question of effectiveness.Well-designed studies focusing on children in particular are needed.

In addition to these reviews, there are a few noteworthy studies that have been published in the last few years not included in the above reviews.

Two larger studies both looked at one-time massage in mixed diagnoses of cancer patients presenting to an outpatient cancer center, and had no control group. Gentile [25] retrospectively reviewed 572 patents who chose either light Swedish massage (281) or healing touch. They received a 45-min session and were asked their pain scores on a numeric score before and immediately following their treatment. The massage group showed improvements in pain scores immediately after massage: pre-treatment pain scores were 4.4 (SD 2.2) and post-treatment pain scores were 2.0 (SD 1.8) *p* < 0.0001, with a difference of −2.5 (SD 1.7). Lopez [26] offered Swedish massage by trained massage therapists to 343 patients and used the Edmonton Symptom Assessment Scale (ESAS; 0–10 scale) pre- and post-massage. Patients could choose between a 30- and 60-min massage. Clinically and statistically significant decreases were found in many symptom areas immediately following massage (pain, fatigue, anxiety, sleep, global distress, physical distress, psychological distress), but only about half of patients completed their post-massage assessments. Of note, there was no observed difference in effects between 30- and 60-min massage duration. While both studies were significantly limited by the lack of consistency in completion of outcome measures and pre–/post designs, they had the benefit of large sample sizes that were able to demonstrate immediate effects on symptoms. Additionally, though in the latter study duration of massage was not randomized, it is notable that longer massage did not necessarily result in better symptom control.

Two studies looked at longer duration of massage in patients with leukemia. Taylor [27] randomized 20 patients to Swedish massage by licensed therapists versus standard of care control. Patients were given 50-min massages three times per week for 7 weeks. Pre–/post massage assessments were done in regard to comfort, relaxation, and stress by numeric scales; patient-reported outcome measures for QOL, anxiety, and pain were performed weekly. Increases in comfort and relaxation and decreases in stress were noted immediately following massage. Additionally, the intervention group had a significant decrease in stress over time, but no significant changes in QOL measures. Miladinia [28] randomized 60 patients with acute leukemia (on active treatment, at least 3 months from diagnosis, and with complaints of pain, fatigue, or sleep ratings) to slow stroke back massage by oncology nurses trained in massage or attention control. Massages lasted 10 min and occurred three times per week for 4 weeks. Pain, fatigue, and sleep disorder intensities were measured using the numeric rating scale weekly over 5 weeks. Sleep quality was measured using the Pittsburgh Sleep Quality Index at baseline and 2 days after study completion. For patients receiving the intervention, symptom scores decreased each week until after massages stopped at which point there was a slight increase, whereas scores for the control group were stable to slightly increased. Additionally, sleep quality improved in the intervention compared with control from baseline to the end of the study. Of note, a barrier to receiving massages in the latter study was that patients were not permitted to get massages when their platelets were less than 15,000, which may not be a necessary restriction. These studies demonstrated that:Massage studies can be performed over a longer period of time, and the effects over time may increase.The intervention may not need to be long to be effective, as the 10-min intervention seemed to be helpful for symptom management.It is possible to demonstrate an effect of massage on sleep, an area often commented on qualitatively but rarely shown in outcome measures.In leukemia patients it is important to consider whether blood counts will be a restriction to massage, as this may prove a barrier.

Izgu [29] evaluated patients receiving oxaliplatin chemotherapy, which has the side effect of causing neuropathic pain. Forty-six patients were randomized to aromatherapy massage of the hands and feet versus standard of care. Patients in the intervention group received 40-min massages three times per week at home for weeks 1–6. The Douleur Neuropathique 4 questionnaire, the numeric pain scale (NPS), and the Piper fatigue scale were measured at weeks 2, 4, 5, and 8. Significant differences were seen in Douleur at week 6, the NPS weeks 2–6, and fatigue at week 8. This study was notable for demonstrating that hand and foot massage may play a role for neuropathic pain specifically, and that fatigue may take longer than pain to see an effect.

Kinkead [30] evaluated breast cancer survivors 3 months to 4 years after treatment with cancer-related fatigue. Sixty-six patients were randomized to Swedish massage by licensed therapists versus an active control (“light touch”) versus a waitlist control. Sessions lasted 45 min and occurred weekly for 6 weeks. Outcome measures included the Multidimensional Fatigue Inventory (MFI) and the PROMIS (Patient-Reported Outcomes Measurement Information System) fatigue at baseline, visits 3 and 6 (post-intervention). The study saw a statistically significant decrease in MFI in the massage and light touch arms, with a corresponding increase for the waitlist control over time. For the PROMIS fatigue measure, both massage and active control had a statistically significant decrease in fatigue (greater improvements in massage vs. light touch), with stable numbers in the waitlist control. This study demonstrated that weekly massage over an extended period of time may be effective for fatigue, that light touch also has some therapeutic benefit that may make it an ineffective control measure, and that the PROMIS measures, which are outcome measures put together by the NIH (National Institutes of Health) to standardize patient reported outcomes, may be an effective tool for measuring symptoms over more extended time periods in massage studies.

## 3. Massage in Children

Beider and Moyer [31] published a review of massage in children about 10 years ago. The authors found 24 randomized controlled trials to evaluate. They noted that more than half did not report sufficient data to permit effect size calculation. However, the authors were able to find that massage therapy decreased anxiety immediately after massage, and that this effect appeared to be greater following a second session of massage. There appeared to be a trend towards improvement in mood, but it did not reach statistical significance, perhaps suggesting that the tools were not sensitive enough or the effect, if there is one, may be subtle and may require increased numbers to have enough power to detect a difference [31].

In pediatric oncology patients, Post-White et al. [32] performed a study with 23 children aged 1–18 years receiving two identical cycles of chemotherapy. Children and a parent were randomized to gentle full body massage versus quiet time (with a “do not disturb” sign on the door) weekly for 4 weeks, and were then crossed over to the other condition for the second cycle of chemotherapy. Physiological measures (heart rate, respiratory rate, and blood pressure) as well as patient-/parent-reported measures (pain, nausea, anxiety) and salivary cortisol were collected before and after massage/quiet time, and fatigue was measured at weeks 1 and 4. Parents also received massages and also completed PRO measures. The study revealed positive statistically significant effects on reducing heart rate and anxiety in the massage group compared to the quiet time control immediately following treatment, and participants had very positive evaluations of their massage experience, citing feeling more relaxed and calm. The effect on anxiety seemed to increase as the weeks progressed. There were no differences in the other measures. The study was limited by including a small sample of children with a large age range, where the outcome measures are hard to compare and the effects may be very different [32].

A few studies have investigated massage in children undergoing bone marrow transplantation. This population is a good one to focus on because they have multiple symptoms in need of improvement, and they are hospitalized for a long period of time. However, they are also challenging because the subjects are very ill, and the numbers are very small. 

Phipps et al. [33] completed two studies. In the first, 50 children with cancer undergoing bone marrow transplantation received professional massage, parent massage, or were standard of care controls. Results indicated statistically significant differences in days to engraftment in the combined (parent and professional) massage group. Results from the professional massage group yielded a significant decrease in anxiety and discomfort immediately following massage [33]. A follow-up study at the same center compared a child intervention group (humor + massage), a parent intervention group (massage + relaxation), and standard of care control and found no differences in depression, quality of life, or post-traumatic stress, though all groups (intervention and control) improved over time and all groups had very high adjustment [34]. These studies highlight the need to be careful about the outcome measures picked to ensure that they might be able to demonstrate the differences the researcher seeks to measure.

Ackerman and Mehling [35,36] randomized stem cell transplant patients 5–18 years old 2:1 to Swedish massage combined with acupressure for 20–30-min sessions from professional therapists three times weekly versus standard of care control. The intervention group also received massage training for parents. The study lasted the entire hospitalization (median 37 days). The outcomes included interviews with parents and providers, patient-reported measures of mood, quality of life, and emotions every 2 weeks, parent outcome measures, and analysis of nursing notes. The study showed qualitative improvements in pain, nausea, relaxation, and sleep and described high levels of parent/caregiver satisfaction with the intervention [35]. Based on nurses’ notes, they found fewer days of mucositis (ES (effect size) 0.63) and lower symptom burden (ES 0.26); based on PROs, they found a trend toward improved fatigue, ES = 0.86, *p* = 0.08, fewer moderate/severe symptoms in a summary measure of fatigue, pain, and nausea (ES = 0.62), and decreased pain (ES = 0.42). The authors noted the need for flexibility in the approach to the intervention to allow kids a sense of control. The effects on fatigue and mucositis may highlight the possible effects of massage on decreasing inflammation, which is particularly important in this population [36].

Jacobs compared 30-min full body massage by licensed therapists in hospitalized adolescents versus standard of care waitlist control for 3 nights, and assessed sleep by actigraphy, as well as fatigue, mood, and anxiety by PRO. There was a statistically significant small increase in number of long sleep episodes (intervention group increased from 11 to 12, while control group decreased from 13 to 11, *p* = 0.049). Additionally, there were increases in mean night-time and overall sleep in the intervention group compared with standard of care, although these differences only approached significance, as well as extremely positive participant and parent feedback on the intervention, particularly for improvements in sleep [37]. This study importantly used an objective measure of sleep in conjunction with PROs, and may suggest a more objective method of measuring massage outcomes.

## 4. Directions for Research

The existent literature on massage in children and adults with cancer highlights several points about what we have learned about the role of massage, the problems that have occurred with research, and many questions about how to best research the topic. Please see Table 1 for a description of approaches for researchers to take.

What we have learned about massage and how to study it:Qualitative reports on the effects of massage are consistently positive.Massage appears to have effects on multiple symptoms, notably pain and anxiety, immediately following massages.Some effects, such as quality of life and fatigue, seem to require a longer duration study to demonstrate differences.Studies need to have sufficient statistical power.

There remain many aspects of the ideal ways to study massage that we still do not know, which makes designing studies difficult.
Dosing: We do not know the optimal dosing of massage, including the best duration of massage for symptom control, the best frequency, or the ideal length of study to see maximum effect. However, some of the studies suggest that massages do not need to be long (i.e., 10–30 min is likely sufficient) to be effective, and that many effects are not long-lasting, so that more frequent dosing may be needed to have more lasting effects. Additionally, since some symptoms such as fatigue may take longer to be ameliorated, longer studies may be required to demonstrate effects.Controls: We do not know the best control arm. Active controls such as light touch may be too effective to use as controls, and attention controls can prove difficult to actualize if patients are not interested in having someone just spend time with them. We must find appropriate controls—“quiet time” controls for hospitalized patients may provide the right balance.Type of massage: several different types of massage and practitioners have been used. While we do not know the ideal approach or type of practitioner, it is important to standardize the approach as much as possible within a study. It is not enough to just study “massage;” the specific techniques being studied have to be well described and reproducible. Larsen describes a standardized massage protocol that should be used as a role model for studies done on the effects of massage [38]. Using licensed massage therapists with research experience and describing the method is important for reproducibility and for accuracy. Additionally, the massage approach may differ depending on the outcome the study is geared at—hand and foot massages may be ideal for neuropathy but not for relieving anxiety, for example.Outcome measures: It is extremely important to be thoughtful about the choice of outcome measures both to be able to compare studies and to be able to adequately demonstrate the effectiveness of massage. Using a combination of shorter-term outcome measures, such as visual analogue scales or numeric rating scales, and longer-term measures, such as the standardized PROMIS measures, may be the best way of demonstrating the short-term and long-term benefits of massage. Additionally, some changes that massage may affect may be quite subtle, and outcome measures need to be sensitive enough to demonstrate subtle but clinically meaningful differences.Patient selection: So many studies are limited by lack of power owing to the study’s small size, so having studies that are broadly inclusive is important, yet several studies are limited by a patient population that is too diverse to draw meaningful conclusions. In pediatrics, there appears to be differences in symptomatology based on ages, so limiting by age is important. In order to increase numbers while assuring some uniformity in patient experience, including patients with known symptoms (i.e., pain, fatigue, anxiety) may be an effective way to target the intervention and show the greatest benefit.Feasibility issues: Studying massage is challenging because of a range of feasibility issues that do not come up with trials that do not study mind–body interventions. Most hospitals and clinics do not have licensed massage therapists with research experience at their disposal, so researchers must consider where they will find the therapists, how they will be incorporated into the clinical setting, and how they will be trained for the study.Statistical issues: So many of the studies that have been done have lacked sufficient power to allow for meaningful conclusions. In addition to choosing outcome measures that are capable of demonstrating the effects of massage, researchers must be sure that studies have enough patients so that a true difference will not be missed due to insufficient power.

## 5. Conclusions

Many prior studies have been done researching the role of massage in children and adults with cancer, but many have had design flaws that limit the usefulness of the findings. This review focused on the major finding of massage, as well as the limitations and challenges in the research that has been done. Based on the findings and limitations of previous work, numerous suggestions have been made for future studies, including thoughts about appropriate dosing, control groups, type of intervention, outcome measures, patient selection, feasibility, and statistics.

## Figures and Tables

**Table 1 children-06-00013-t001:** Massage study planning table. PROs: patient-reported outcome measures; QOL: quality of life.

Variable	Considerations	Recommendations
Dose	Length of session, frequency, type of massage	10–30 mins, trained therapists, ideal frequency and length of study unknown
Control	Light touch or alternative modality not sufficiently different or unacceptable to patients	Randomized control studies required; quiet time a valid control
Massage	Multiple types, variation between providers and between sessions	Use one modality, a standard approach with limited flexibility
Outcomes	Expected onset and magnitude of change, PROs, QoL, clinical, proxy, language, focus on few outcomes (e.g., pain, sleep, fatigue, anxiety), blinded data collection	Immediate pre/post—pain-VAS (visual analogue scale), clinical (BP, HR (blood pressure, heart rate)) Frequent (daily)—sleep diary/actigraphy, nausea Weekly—QoL, PRO (e.g., PROMIS) End of study—perceived patient/clinician burden benefit
Patient	Number, age groups, symptom burden/treatment	Statistically significant groups of like patients, stratify by age and treatment (perhaps gender)
Feasibility	Massage therapists with research/clinical training, access to patients, hospital policy, consistent sessions, proxy presence, waitlist controls	Require hospital/administration backing, enthusiastic therapists ready to be research trained
Statistics	Number of patients in each arm, strata, realistic study participation projection for timeframe	Sufficient power essential

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
