# Peer review of "The Power of Massage in Children with Cancer—How Can We Do Effective Research?"

_children, 2019, doi:10.3390/children6010013_

Round 1

Reviewer 1 Report

The authors provided a thoughtful & engaging review of an important topic; appreciate the paper's call to improved rigor & consistency in integrative therapy reporting.

Very minor typos on Page 2... The word "to" is missing in Line 74 ("massage appears TO have") & then in the final line of page 2, I believe the semicolon after "included" should perhaps be a colon.

1. Please consider adding a table to highlight the text features of Page 6, Line 274 onward... A table that defines best practice for dosing, controls, type of massage reporting and examples of strongly completed work (the references of good examples are already cited in the paragraphs... would welcome this also in table format so the reader can have definitions, examples, and maybe a summary column of recommendation for reporting that domain)

2. Some of the "Recommendations" summarized from the systematic reviews are statements rather than recommendations.... For all areas of the paper that have numeric recommendations would encourage consistent language (appreciated the recs that started with a verb or actionable phrase)

3. Consider recommending that the child's and/family's perceived benefit/burden of research participation be part of massage reporting best practice 

4. Maybe add a few non-research recommendations as best practice such as all research reporting on massage should report on consent but ALSO unique  to massage therapy should report on the language used to consistently obtain "permission to touch from child or family" (added consent not just to be part of research but to be part of massage).

Great paper! Adds thoughtfully to the literature.

Author Response

The authors provided a thoughtful & engaging review of an important topic; appreciate the paper's call to improved rigor & consistency in integrative therapy reporting.

Thank you

Very minor typos on Page 2... The word "to" is missing in Line 74 ("massage appears TO have") & then in the final line of page 2, I believe the semicolon after "included" should perhaps be a colon.

These were fixed

1. Please consider adding a table to highlight the text features of Page 6, Line 274 onward... A table that defines best practice for dosing, controls, type of massage reporting and examples of strongly completed work (the references of good examples are already cited in the paragraphs... would welcome this also in table format so the reader can have definitions, examples, and maybe a summary column of recommendation for reporting that domain)

See massage study planning table, as yet unable to offer dose guidance

2. Some of the "Recommendations" summarized from the systematic reviews are statements rather than recommendations.... For all areas of the paper that have numeric recommendations would encourage consistent language (appreciated the recs that started with a verb or actionable phrase)

The recommendation sections of the review were looked at carefully and where appropriate language was changed

3. Consider recommending that the child's and/family's perceived benefit/burden of research participation be part of massage reporting best practice 

Added to table

 4. Maybe add a few non-research recommendations as best practice such as all research reporting on massage should report on consent but ALSO unique  to massage therapy should report on the language used to consistently obtain "permission to touch from child or family" (added consent not just to be part of research but to be part of massage).

Interesting thought, but beyond the scope of this review

Great paper! Adds thoughtfully to the literature.

Reviewer 2 Report

Thank you for this article to highlight massage as an integrative modality for children with cancer.

Also very much appreciate the guidance for future research studies on massage. 

Introduction:

After the first sentence, it may be important to get your objective for the manuscript across right there to guide readers about the purpose of the manuscript. For example, "Children and adolescents with cancer experience multiple troubling and often inter-related 25 symptoms, including disturbed sleep, fatigue, anxiety, and pain. The purpose of this manuscript is to summarize/outline the evidence regarding massage therapy for children with cancer."

 the last sentence of the introduction "This report aims to summarize what research has shown us about massage, and to outline some of the challenges present in researching mind-body techniques like massage and some of the goals for future research." Can be changed to "In addition to summarizing the research regarding massage, this manuscript will also outline the challenges to researching massage therapy, as well as guiding next steps for future research." 

line 57-58: may change "Unfortunately, many of the studies have been underpowered, or with other design flaws that make them difficult to interpret." to "unfortunately, many of these studies have been underpowered or have design flaws that make them difficult to interpret."

line 78: Change "Other outcomes, such as sleep and quality of life were more inconsistent." to add comma after quality of life. 

line 105: need space between literature and 1990 "After reviewing 105 all literature1990 to 2013, 12 studies were included"

Line 214 - PRO measures - is this acronym for patient reported outcomes? I didn't see the acronym spelled out above this, but if it is then no need to spell out patient reported outcomes here.

Author Response

Thank you for this article to highlight massage as an integrative modality for children with cancer.

Also very much appreciate the guidance for future research studies on massage. 

Thank you

Introduction:

After the first sentence, it may be important to get your objective for the manuscript across right there to guide readers about the purpose of the manuscript. For example, "Children and adolescents with cancer experience multiple troubling and often inter-related 25 symptoms, including disturbed sleep, fatigue, anxiety, and pain. The purpose of this manuscript is to summarize/outline the evidence regarding massage therapy for children with cancer."

Good suggestion, this was added

The last sentence of the introduction "This report aims to summarize what research has shown us about massage, and to outline some of the challenges present in researching mind-body techniques like massage and some of the goals for future research." Can be changed to "In addition to summarizing the research regarding massage, this manuscript will also outline the challenges to researching massage therapy, as well as guiding next steps for future research." 

This was changed

Line 57-58: may change "Unfortunately, many of the studies have been underpowered, or with other design flaws that make them difficult to interpret." to "unfortunately, many of these studies have been underpowered or have design flaws that make them difficult to interpret."

fixed

Line 78: Change "Other outcomes, such as sleep and quality of life were more inconsistent." to add comma after quality of life. 

fixed

Line 105: need space between literature and 1990 "After reviewing all literature1990 to 2013, 12 studies were included"

fixed

Line 214 - PRO measures - is this acronym for patient reported outcomes? I didn't see the acronym spelled out above this, but if it is then no need to spell out patient reported outcomes here.

fixed

Reviewer 3 Report

An urgent need for the manuscript is the description of the search methodology used by authors for selecting the papers to assemble this review: databases consulted, keywords used, inclusion criteria, avoiding bias, etc. This action requires adding a couple of paragraphs.

Some minor corrections:

- In Abstract, replace "pub med" by "PubMed (database of US National Library of Medicine, National Institutes of Health)"

- The final statements Author Contributions (line 325) and Funding (line 332) are is a draft template form.

Author Response

An urgent need for the manuscript is the description of the search methodology used by authors for selecting the papers to assemble this review: databases consulted, keywords used, inclusion criteria, avoiding bias, etc. This action requires adding a couple of paragraphs.

Description of approach added

Some minor corrections:

In Abstract, replace "pub med" by "PubMed (database of US National Library of Medicine, National Institutes of Health)"

changed

The final statements Author Contributions (line 325) and Funding (line 332) are is a draft template form.

Changed